# Lack of Oestrogen Receptor Expression in Breast Cancer Cells Does Not Correlate with Kisspeptin Signalling and Migration

**DOI:** 10.3390/ijms23158744

**Published:** 2022-08-06

**Authors:** Udochi F. Azubuike, Claire L. Newton, Iman van den Bout

**Affiliations:** 1Department of Physiology, Faculty of Health Sciences, University of Pretoria, Private Bag X323, Pretoria 0031, South Africa; 2Centre for Neuroendocrinology, Department of Immunology, Faculty of Health Sciences, University of Pretoria, Private Bag X323, Pretoria 0031, South Africa; 3Department of Immunology, Faculty of Health Sciences, University of Pretoria, Private Bag X323, Pretoria 0031, South Africa

**Keywords:** kisspeptin, KISS1R, metastasis, breast cancer, oestrogen receptor, calcium signalling, ERK, cell migration

## Abstract

Kisspeptin is an anti-metastatic mediator in many cancer types, acting through its receptor, KISS1R. However, controversy remains regarding its role in breast cancer since both pro- and anti-metastatic roles have been ascribed to it. In KISS1R overexpressing triple-negative breast cancer (TNBC) cells, stimulation has been associated with increased invasion and MMP-9 expression, leading to the suggestion that hormone receptor status determines the metastatic effects of kisspeptin. To assess the veracity of this claim, we compared endogenous KISS1R signalling and physiological output in the hormone receptor-negative MDA-MB-231 and BT-20 cell lines after KP-10 (shortest active kisspeptin peptide) stimulation. MDA-MB-231 cells are metastatic when implanted in mice while BT-20 are not and remain epithelial-like. We show that both cell lines express *KISS1R* mRNA and respond to KP-10 by elevating calcium mobilisation. However, KP-10 stimulation induced migration of MDA-MB-231, but not BT-20 cells, in a calcium-dependent manner. Moreover, only BT-20 cells responded to KP-10 by increasing ERK phosphorylation in a β-arrestin-dependent manner. Interestingly, both cell lines displayed different complements of β-arrestin 1 and 2 expression. Overall, our data shows that, in TNBC, it is not universally true that kisspeptin/KISS1R stimulate migration or pro-metastatic behaviour, as divergent responses were observed in the two TNBC lines tested. Whether this divergence is related to the observed differences in β-arrestin complements warrants further investigation and may enable further stratification of the ability of kisspeptin to influence breast tumour behaviour.

## 1. Introduction

Kisspeptin was first shown to inhibit metastasis in melanoma in 1996 [1]. The initial product of the *KISS1* gene, prepro-kisspeptin, is proteolytically cleaved to produce shorter peptides (collectively termed kisspeptin): kisspeptin-54, kisspeptin-14, kisspeptin-13, and kisspeptin-10 (KP-10) [2,3], with the common 10 amino acids at the carboxyl terminal (KP-10) being essential for activation of the cognate receptor, KISS1R [3]. KISS1R is a G protein-coupled receptor, activation of which primarily results in signal transduction via Gα_q/11_ G proteins, resulting in activation of phospholipase C (PLC) and downstream calcium mobilization [4]. Activation of KISS1R also results in the activation of extracellular regulated kinase (ERK 1/2), p38 mitogen-activated protein kinase (p38MAPK) [3], and activation of the Akt pathway [4]. Moreover, KISS1R activation can impact matrix metalloproteinase 9 (MMP9) expression and activation, influence actin reorganization of stress fibres, and inhibit cell movement in different cell contexts [5,6].

Kisspeptin/KISS1R has been shown to suppress metastasis in many cancer types, including melanoma [1], colorectal cancer [7], bladder cancer [8], lung cancer [9], prostate cancer [10], ovarian cancer [11], and pancreatic cancer [12]. In these cancers, KISS1R signalling suppresses metastasis by repressing MMP-9 activity [7], thereby inhibiting cancer cell invasion and migration [7].

Despite its clear anti-metastatic effects in many cancers, in breast cancer, the role of kisspeptin/KISS1R is still not clearly understood. Studies by Stark and colleagues [13] have shown that, in breast cancer patients, *KISS1* mRNA expression is lower in brain metastases compared to the primary tumours. However, studies by Martin et al. [14], Cvetkovic et al. [15], Goertzen et al. [16], and Blake et al. [17] have shown that *KISS1* and/or *KISS1R* mRNA/protein is/are highly expressed in metastatic breast cancer cells compared to primary tumours. In addition, some studies have indicated that high expression levels also correlate with aggressive behaviour, such as increased invasion and migration, of metastatic breast cancer cells in vitro [5,16], while contrasting studies have suggested that kisspeptin can inhibit cell growth and migration in vitro [18,19]. Interestingly, previous studies have also suggested that the differential effects on kisspeptin may be related to oestrogen receptor expression. In hormone receptor-negative breast cancer cell lines, activation of KISS1R has been shown to promote epithelial-to-mesenchymal transition (EMT) and invadopodia formation along with increased migration and invasion [5,16]. A direct comparison between oestrogen receptor-negative (ER−) and oestrogen receptor-positive (ER+) cells showed that kisspeptin had opposing effects on invasion and migration, with the ER+ cells being inhibited while the ER− cells were stimulated [15]. These studies point to a possible stratification of breast cancer with regards to the effect of kisspeptin into an oestrogen receptor-positive group where kisspeptin acts in an anti-metastatic manner, and an oestrogen receptor-negative group where it acts in a pro-metastatic manner. However, many of the in vitro studies using cell lines employ recombinant lines exogenously overexpressing KISS1R, making these cells exquisitely sensitive to kisspeptin and thus making such firm conclusions difficult [20]. Indeed, it has been demonstrated that in several breast cancer cell lines the anti-proliferative effects observed after kisspeptin stimulation is dependent on exogenous overexpression of the receptor KISS1R [20].

Within the triple-negative subtype of breast cancer, a number of cell lines exist that are widely used for preclinical investigation. In this study, we selected two such lines (the metastatic MDA-MB-231 cell line and the non-metastatic BT-20 cell line) to determine if the signalling observed downstream from endogenous KISS1R is similar, even if the cell lines have opposing metastatic and migratory characteristics. With such data we aimed to verify whether the lack of oestrogen receptor expression does universally transform kisspeptin-dependent signalling from exerting an anti-metastatic to a pro-metastatic effect in breast cancer cells. KP-10-mediated calcium mobilization, cell migration, ERK signalling, and cell proliferation were examined. The results show that KP-10 induces differential signalling in the two cell lines. While both cell lines responded to KP-10 by mobilizing calcium, only MDA-MB-231 cells responded by increasing cell migration, dependent on calcium mobilization. Only BT-20 cells exhibited β-arrestin-dependent ERK1/2 activation after KP-10 stimulation. KP-10 had no effect on the proliferation of either cell line. Taken together, we show that not all oestrogen receptor-negative breast cancer cells increase migration after KP-10 stimulation even if they generate an ERK response or mobilize calcium after stimulation. Thus, stratification by oestrogen receptor does not necessarily predict a pro-metastatic effect of kisspeptin and additional cellular context differences must play a role in directing such a response.

## 2. Results

### 2.1. KISS1R Expression

To assess how KP-10 and KISS1R affect cell signalling and migration of triple-negative breast cancer cells, appropriate cell lines that expressed endogenous KISS1R were identified by analysing the mRNA levels using RT-PCR (Figure 1). Total RNA of cells was isolated, and cDNA generated before gene-specific primers were used to amplify *KISS1R*. MDA-MB-231 and BT-20 cells were found to express similar levels of *KISS1R* mRNA. Endogenous KISS1R protein expression was also assessed by Western blot (data not shown) using both a custom-generated and commercially available antibody, but control experiments were unable to produce convincing evidence that either specifically detected KISS1R. Therefore, we chose to assess KP-10-dependent signalling to confirm expression of functional KISS1R in these cell lines.

### 2.2. Calcium Mobilization

Calcium mobilization is activated downstream of GPCRs via Gα_q_, which induces phospholipase C (PLC) to generate diacylglycerol (DAG) and inositol triphosphate (IP_3_). IP_3_ stimulates IP_3_ receptors (IP_3_Rs) on the endoplasmic reticulum to release calcium [4]. Therefore, to assess the functionality of KISS1R in the different cell lines, calcium mobilization was measured using a calcium probe, Fluo 3 AM, after stimulation with KP-10. Cells were cultured overnight in 8-well microslide plates. The next day, cells were stained with Fluo-3 AM and imaged for fluorescence intensity before and after stimulation with 100 nM KP-10 (a concentration commonly utilized in examining kisspeptin responses that has been shown to induce MDA-MB-231 cell migration in previous studies [5,15,21]) (Figure 2). As a control, cells were treated with 1 µM Ionomycin (a calcium mobiliser) and 1 M CaCl_2_. Stimulation with KP-10 elicited a rapid and significant increase in intracellular calcium in both the MDA-MB-231 and BT-20 cell lines. BT-20 cells reached an average maximum level of calcium of 25.2% of that elicited in the positive control after 61.5 s, while MDA-MB-231 cells reached a maximum of 45%, 45.5 s after stimulation. However, there was some cell-to-cell variability and, overall, there was no significant difference in amplitude between the cell lines. There was also no difference in time to maximum amplitude. Thus, stimulation with KP-10 results in calcium mobilization in both cell lines, indicating that they possess endogenous functional KISS1R to mediate this response.

### 2.3. Cell Migration

Previous studies have suggested that kisspeptin enhances migration in ER-negative breast cancer cell lines and inhibits migration in ER-positive cell lines [22]. To assess if KP-10 stimulation influences cell migration, in vitro wound-healing assays of confluent BT-20 and MDA-MB-231 (Figure 3a) cells were performed over 18 h following KP-10 exposure (Figure 3). BT-20 cells (Figure 3b) did not migrate under serum-free conditions and the addition of KP-10 did not have any additional effect on their migration. In contrast, KP-10 stimulation of MDA-MB-231 cells increased migration by approximately 10% compared to the vehicle control (Figure 3c). Subsequently, the effect of KP-10 on cell migration in the MDA-MB-231 cells was verified using an Oris™ migration assay kit (Figure 3d), which confirmed that KP-10 increased the migration MDA-MB-231 cells (by approximately 30% compared to the vehicle control).

Since calcium mobilization has been shown to influence cell migration by regulating cytoskeletal organization, it was hypothesized that the KP-10-induced increases in migration of MDA-MB-231 cells may be calcium dependent. To test this, calcium mobilization was inhibited using the Ca^2+^ chelator, BAPTA-AM, prior to stimulation with KP-10 and measurement of migration (Figure 3e). Interestingly, the KP-10-induced increase in migration was absent when the cells were pre-treated with 10 µM BAPTA-AM. Thus, KP-10 stimulates migration in a calcium-dependent manner in the MDA-MB-231 cell line but similar calcium mobilization in the BT-20 cell line does not result in altered migration rates.

### 2.4. ERK Activation

#### 2.4.1. ERK Activation Occurs Only in BT-20 Cells

ERK is a central signal transduction component activated by a number of different inputs, including G protein-dependent and -independent GPCR signalling. Previous studies have demonstrated that ERK activation occurs downstream of KISS1R [21,23] and its activation has been linked to mitogenic effects in many cell lines. Therefore, to explore the potential role of ERK activation in the differential effects of KP-10 on the two cell lines, its activation in response to KP-10 stimulation was assessed. Cells were stimulated with 100 nM KP-10. Cells were first serum deprived for 4 h before being stimulated with KP-10 for the indicated time intervals up to 60 min after which lysates were made and phosphorylated ERK and total ERK levels were visualized by Western blot (Figure 4). Statistical analysis of multiple biological repeats indicates that KP-10 induced ERK phosphorylation at 60 min in the BT-20 cells. In the MDA-MB-231 cells, there was a small increase in ERK phosphorylation from 10 min, but this was not statistically significant.

#### 2.4.2. ERK Activation in BT-20 Cells Is Dependent on β-Arrestin

While GPCRs can activate ERK through their associated G proteins, leading to rapid activation, G protein-independent responses also occur whereby β-arrestin is recruited to the GPCR and a slower pathway to ERK activation is initiated [24]. Since ERK activation in the BT-20 cells was only observed at 60 min, it was hypothesized that this activation is dependent on β-arrestin. To test this, ERK activation in BT-20 cells was assessed after KP-10 stimulation in the presence or absence of the arrestin inhibitor, Barbadin. BT-20 cells were exposed to Barbadin at different concentrations for 30 min before stimulation for 60 min with 100 nM KP-10 (Figure 5). Barbadin effectively inhibited KP-10-induced ERK 1/2 phosphorylation, suggesting that this response was indeed β-arrestin mediated.

#### 2.4.3. Cell Lines Differentially Express β-Arrestin Isoforms

Since BT-20, but not MDA-MB-231, cells activated ERK, and this stimulation was β-arrestin dependent, it was postulated that this could be linked to differences in β-arrestin expression in the two cell lines. Indeed, the two different β-arrestins (β-arrestin 1 and 2) have been linked to different effects downstream of GPCRs in different cell types; in some cases exerting opposing effects on ERK activation while acting synergistically in others [25]. Therefore, β-arrestin 1 and 2 expression were examined by Western blot in the two cell lines (Figure 6). BT-20 cells expressed more than 5-fold higher levels of β-arrestin 1 compared to MDA-MB-231 cells in which its expression was barely detectable. Conversely, β-arrestin 2 was expressed 3.5-fold higher in the MDA-MB-231 cells than the BT-20 cells. These data suggest that there is divergence in arrestin isoform expression between the two cell lines, which may explain the lack of ERK activation in the MDA-MB-231 cells compared to the BT-20 cells.

### 2.5. Cell Proliferation

In addition to effects on invasion and migration, several studies have shown that kisspeptin can affect cell proliferation. Again, contrasting effects have been shown in different cell lines with a clear divide between ER-negative and ER-positive cell lines [18,19]. To assess if the two ER-negative cell lines adopt different proliferation rates in response to KP-10, cell proliferation was assessed by resazurin conversion assay over 5 days with continuous KP-10 treatment (120 h), (Figure 7). The proliferation rate of MDA-MB-231 cells was significantly higher than the BT-20 cells, even in the absence of serum, as expected. Resazurin conversion rose by ~33% in BT-20 cells over 5 days while in MDA-MB-231 cells it rose by ~100%. However, the addition of KP-10 did not alter the rate of proliferation in either cell line. Thus, KP-10 has no effect on proliferation despite differences in ERK activation after stimulation in BT-20 cells.

## 3. Discussion

While it seems clear that kisspeptin and its receptor KISS1R play a role in preventing or inhibiting metastasis in a number of cancer types, controversy remains regarding its role in breast cancer. Contrasting studies have shown either pro- or anti-metastatic roles for this peptide and its receptor in breast cancer. For instance, *KISS1* expression was shown to be elevated in metastatic breast tumours in several studies [14,17,26], while another showed the opposite [13]. Furthermore, in vitro studies have also resulted in contrasting conclusions regarding kisspeptin and KISS1R function in breast cancer, with some suggesting they act in a pro-metastatic manner while others suggest they act in an anti-metastatic manner [5,15,19]. In fact, it has been shown that overexpression of KISS1R itself influences the way cells respond and thus studies based on cell lines overexpressing exogenous KISS1R should be treated with caution [20]. Overall, a theory has been proposed that the subtype of breast cancer may determine the effects kisspeptin and KISS1R have on metastasis. Oestrogen receptor-positive breast cancer cells seem to respond to kisspeptin by reducing migration and proliferation. In contrast, several oestrogen receptor-negative cell lines respond to kisspeptin by increasing migration, invasion, and proliferation [15]. Thus, this distinction of oestrogen receptor status could be the source of the abovementioned controversy. However, this would imply that all oestrogen receptor-negative breast cancer cell lines respond to kisspeptin through the induction of similar intracellular signalling, resulting in increased cell proliferation/migration. Herein, we sought to test whether this distinction holds true. We chose to analyse the response to KP-10 stimulation of two TNBC lines with differing cell behaviours. MDA-MB-231 cells have previously been shown to respond to kisspeptin by increasing cell migration [15]. These cells were isolated from a metastatic tumour and when injected into mice will readily form new metastatic tumours [27]. In contrast, BT-20 cells, while also triple negative, were isolated from a primary tumour and do not form metastatic tumours when injected into mice nor do they migrate in vitro [28]. To determine if the current perception that kisspeptin response in breast cancer can be stratified by oestrogen receptor is correct, we hypothesized that these two divergent TNBC lines should behave similarly in response to KP-10. Importantly, no exogenous KISS1R expression was introduced to the cells so that only endogenous KISS1R responses could be assessed.

To verify that both cell lines express KISS1R and thus would be able to respond to KP-10, receptor expression was determined by RT-PCR analysis and it was found that *KISS1R* mRNA was present at a similar level in both cell lines. Unfortunately, a lack of reliable antibodies prevented assessment of endogenous KISS1R protein expression. Thus, another method was adopted to determine if endogenous and functional KISS1R was expressed by the cell lines. As KISS1R signals primarily via coupling to Gα_q/11_ G proteins, the activation of which triggers a pathway that results in calcium mobilization [3], to determine if functional KISS1R was expressed, calcium mobilization was measured after KP-10 stimulation. Calcium mobilization after stimulation with KP-10 increased rapidly in both cell lines. No statistically significant differences in the amplitudes of maximum mobilization and time to maximum mobilization were measured between the cell lines. These data indicate that KP-10 can activate endogenous KISS1R similarly in both cell lines.

Previous studies have shown that 100 nM KP-10 induces the migration of MDA-MB-231 cells [5,16]. Similarly, we show here that MDA-MB-231 cell migration is increased after KP-10 exposure. In contrast, BT-20 cells are non-migratory and exposure to KP-10 did not overcome this characteristic of the cells. This suggests that there are differences in response to KP-10 among TNBC lines.

Interestingly, the effect of KP-10 on MDA-MB-231 migration is dependent on calcium mobilization since pretreatment with BAPTA-AM abolished any increase in migration. Since both cell lines do mobilize calcium in response to KP-10 stimulation, we must conclude that additional factors must also play a role in determining if KP-10 can induce migration.

In contrast to calcium mobilization, ERK phosphorylation was differentially activated in response to KP-10 stimulation in the two cell lines. Surprisingly, the data indicated that endogenous KISS1R in MDA-MB-231 cells did not activate a pathway leading to ERK phosphorylation. A previous study using MDA-MB-231 cells, showed that there was an increase in ERK phosphorylation at 10 and 30 min after overnight starvation and exposure to 100 nM KP-10 although it is unclear if these increases were statistically significant [23]. Our data do demonstrate a slight but statistically insignificant increase at 10 min. It is important to remember that serum starvation was only carried out for 4 h (since overnight starvation led to poor cell viability); thus, small increases could have been masked. Nonetheless, unlike the MDA-MB-231 cells, ERK phosphorylation was robustly increased after 60 min following KP-10 stimulation in BT-20 cells.

The non-visual arrestins, β-arrestin 1 and 2, when overexpressed, can mediate the internalization of exogenous KISS1R in HEK293 cells [23]. Moreover, it has been shown that knockdown of endogenous β-arrestin 2 can ameliorate ERK phosphorylation after KP-10 exposure in MDA-MB-231 cells [23]. Therefore, we hypothesized that the ERK phosphorylation seen in the BT-20 cells could be β-arrestin dependent. Inhibiting arrestin function by exposing cells to Barbadin completely abrogated ERK phosphorylation induced by KP-10. Barbadin has been identified as a potent inhibitor of the interaction between β-arrestin and the β2-adaptin subunit of the clathrin adaptor protein AP-2 [29]. Not only does this inhibit receptor internalization but it also inhibits arrestin-mediated ERK activation and cAMP production. Thus, it can be concluded that the ERK activation observed in the BT-20 cells is dependent on the coupling of β-arrestin and AP-2. In the abovementioned article, β-arrestin 2 knockdown by siRNA inhibited ERK signalling in MDA-MB-231 cells [23]. Our study shows that BT-20 cells express significantly higher levels of β-arrestin 1 compared to MDA-MB-231 cells while, in contrast, MDA-MB-231 cells express significantly higher levels of β-arrestin 2 than the BT-20 cells. Mouse embryonic fibroblasts isolated from β-arrestin 1 or 2 knockout mice have previously been used to investigate their role in KISS1R signalling. KISS1R was exogenously overexpressed in these cells and ERK activation assessed after KP-10 stimulation [21]. In this setting, loss of β-arrestin 1 led to increased ERK phosphorylation at 5 and 10 min after stimulation, while β-arrestin 2 knockout cells displayed a lack of ERK phosphorylation after stimulation. These data suggest that β-arrestin 2 is the mediator of KISS1R-mediated ERK phosphorylation (at least at early time points), while β-arrestin 1 inhibits this pathway. This somewhat contrasts with our data that show that BT-20 cells have high levels of endogenous β-arrestin 1 and inhibition of β-arrestin (using Barbadin, which inhibits both β-arrestins 1 and 2) inhibits late ERK phosphorylation. It is also important to note that in the model described above only early timepoints in ERK activation were investigated. It is therefore still possible that the different arrestin isoforms may play a role at different times in the ERK pathway. Our data currently shows only a correlation between arrestin isoform expression and ERK activation. Future siRNA or CRISPR experiments removing the arrestins from BT-20 cells would be able to confirm which isoform is responsible for ERK activation.

An interesting study showed that the antiproliferative effects of kisspeptin reported by some studies were mainly due to the exogenous overexpression of KISS1R, while cancer cell lines with endogenous KISS1R expression only did not respond to KP-10 to alter their proliferation rate [20]. In agreement with this, we found that neither the BT-20 nor MDA-MB-231 cell lines responded to KP-10 in proliferation assays. These data suggest that the activation of ERK downstream of KISS1R in BT-20 cells is not linked to cell proliferation.

In conclusion, we present herein evidence that the idea that the oestrogen receptor status of breast cancer can be correlated to the effect of KP-10 on cell signalling and metastatic potential cannot be universally true. While receptor status may play a role, our data show that two different oestrogen receptor-negative cell lines have differential responses to KP-10 stimulation with regards to migration and ERK signalling while calcium mobilization is activated in both. Therefore, additional, as yet unknown, cell-context-specific factors must also play a role in determining the cellular response to kisspeptin.

## 4. Materials and Methods

### 4.1. Chemicals and Materials

Kisspeptin-10 (KP-10, amino acid sequence, Tyr-Asn-Trp-Asn-Ser-Phe-Gly-Leu-Arg-Phe-NH_2_) was synthesized by GL Biochem (Shanghai) Ltd., Shanghai, China. DMEM, DMEM-F12 and foetal calf serum (FCS) were purchased from Invitrogen (Waltham, MA, USA). Originally, KP-10 was solubilized in 0.2% propylene glycol and was used in the first migration assays (scratch assays). Subsequent preparations of KP-10 were prepared in 0.1% DMSO such that, for all assays other than the scratch assay, the vehicle control was 0.1% DMSO. BAPTA-AM and Resazurin were purchased from Sigma-Aldrich (St. Louis, MO, USA), Fluo-3 AM and Ionomycin were purchased from Invitrogen (Waltham, MA, USA), and Barbadin was purchased from Axon Medchem (Groningen, The Netherlands). An Oris™ migration assay kit (consisting of stoppers, stopper remover and detection mask) was purchased from Platypus Technologies (Fitchburg, WI, USA).

### 4.2. RNA Extraction, cDNA Synthesis, RT-PCR

Total RNA isolation was performed using the RNeasy Mini Kit from Qiagen (Hilden, Germany). The concentration of RNA was measured using a NanoDrop™ 2000 (Thermo Scientific, Waltham, MA, USA). cDNA synthesis was performed using 2 µg of the total RNA, 2.5 µM random hexamers, and 1.25 U/µL Superscript III reverse transcriptase from Invitrogen (Waltham, MA, USA), with the following cycling conditions: 25 °C for 10 min, 48 °C for 30 min, 95 °C for 5 min. Real-time qPCR was performed with SYBR using Luna^®^ Universal qPCR Master Mix kit from New England Biolab (Ipswich, MA, USA), following the manufacturer’s protocol, with primers specific for KISS1R (Human KISS1R forward primer, 5′-CAACTTCTACATCGCCAACC-3′ and Human KISS1R reverse primer, 5′-ACATGAAGTCGCCCAGCA-3′).

### 4.3. Cell Culture

The human triple-negative breast cancer cell lines, BT-20 and MDA-MB-231, were purchased from ATCC (Manassas, VA, USA). The MDA-MB-231 cell line was cultured in DMEM (Invitrogen, Waltham, MA, USA) supplemented with 10% (*v/v*) FCS (Complete DMEM), while the BT-20 cell line was cultured in DMEM-F12 (Invitrogen, Waltham, MA, USA) supplemented with 10% (*v/v*) FCS (Complete DMEM-F12). Both were maintained in a humidified incubator at 37 °C with 5% CO_2_ and a 95% relative humidity.

### 4.4. Immunoblotting

To assess the expression of β-arrestin 1 and 2 proteins, BT-20 and MDA-MB-231 cells grown to 80 % confluence were lysed using RIPA buffer (20 mM HEPES, 150 mM NaCl, 1 mM EDTA, 1% NP40 and 0.2% SDS) supplemented with Complete protease inhibitor cocktail (Roche, Basel, Switzerland), and equal amounts (10 µg) of protein were mixed with 4× loading buffer (8% SDS, 20% β-mercaptoethanol, 40% glycerol, 0.008% bromophenol blue, 0.2 M Tris-HCl) and separated by electrophoresis on 4–20% gradient NuPage™ gels from Invitrogen (Waltham, MA, USA) and blotted onto PVDF membranes. Endogenous protein expression was visualized using the following primary antibodies: mouse monoclonal β-arrestin1 (Transduction Laboratories, 1:1000), goat polyclonal β-arrestin 2 (Abcam, 1:1000), mouse monoclonal β-tubulin (Sigma-Aldrich, 1:1000), followed by the secondary antibodies: horseradish conjugated goat-anti-mouse (Bio-Rad, 1:10,000) or Donkey anti-goat Alexa Fluor 555 (Invitrogen, 1:5000) where appropriate. The blots were visualized by chemiluminescence and fluorescence, and densitometric analysis of the bands was performed using Image Lab software from Bio-Rad (Hercules, CA, USA). β-tubulin was used as a loading control. β-arrestin 1 and 2 bands were normalised to β-tubulin in the same lanes. The experiment was repeated at least three times independently.

To assess the expression of phosphorylated and total ERK1/2, BT-20 and MDA-MB-231 cells were grown to 70–80% confluence, serum-starved for 4 h, and then stimulated with 100 nM KP-10 for 0, 5, 10, 30, 45, and 60 min. Thereafter, cells were lysed in RIPA buffer supplemented with Complete protease and PhosStop phosphatase inhibitor cocktails (Roche, Basel, Switzerland). Thereafter the lysates were mixed with 4× loading buffer and denatured by boiling at 95 °C for 5 min and equal amounts (10 µg of the BT-20 and 15 µg of the MDA-MB-231) of the lysates were separated on the 4–20% gradient NuPage™ gels (Invitrogen, Waltham, MA, USA) and transferred to PVDF membranes for immunoblotting. Phosphorylated ERK1/2 and total-ERK1/2 were detected using the primary antibodies rabbit polyclonal phospho-p44/42 MAPK (Cell Signalling, 1:750) and rabbit polyclonal p44/42 MAPK (Cell Signalling, 1:750), followed by the secondary antibody, horseradish peroxidase conjugated goat-anti-rabbit (Bio-Rad, 1: 10,000). The blots were visualised by chemiluminescence and densitometric analysis of the bands was performed using Image Lab software. β-tubulin was used as a loading control. Phospho-ERK1/2 and total-ERK1/2 bands were normalised to β-tubulin in the same lanes. Thereafter, each KP-10 treatment or different concentrations of Barbadin or Barbadin + KP-10 was/were divided by the respective normalised total ERK1/2 signals and then finally divided by the DMSO vehicle control, in order to plot a graph of relative ERK1/2 phosphorylation over time or relative ERK1/2 over the different concentrations of Barbadin. The experiment was repeated at least three times independently.

To assess the effect of β-arrestin 1/2 inhibition on ERK1/2 phosphorylation, BT-20 cells were first pre-treated for 30 min with different concentrations of Barbadin (0.1 µM, 0.5 µM, 1 µM, 2 µM, and 3 µM) before being treated with 100 nM KP-10 for 60 min, lysed and analysed by Western blotting using the ERK1/2 method described above.

### 4.5. Calcium Mobilisation Assay

BT-20 and MDA-MB-231 cells were seeded at a density of 5 × 10^4^ and 4 × 10^4^ cells per well (200 µL per well), respectively, in 8-well microslide plates (Ibidi, Bavaria, Germany) and incubated overnight in a humidified incubator. The next day, the cells were loaded with a dye-loading solution containing 2.5 µM Fluo-3 AM, 0.2 mg/mL of Pluronic™ F-127, and 2.5 mM probenecid in HBSS-BSA buffer (5.4 mM KCl, 0.5 mM MgCl_2_-6H_2_O, 0.4 mM MgSO_4_-7H_2_O, 0.44 mM KH_2_PO_4_, 0.34 mM Na_2_HPO_4_-7H_2_O, 1.3 mM CaCl_2_, 5.5 mM d-glucose, 4.2 mM NaHCO_3_, and 1 mg/mL of BSA), for 30 min at 37 °C in a humidified incubator. Thereafter, the dye-loading solution was removed and replaced with HBSS-BSA buffer for a further 30 min at 37 °C in a humidified incubator. Calcium responses were measured in real time using a confocal microscope every 1.5 s for at least 300 frames, before and after stimulation with 100 nM KP-10 or 1 µM Ionomycin + 1 M CaCl_2_. The data were quantified using ImageJ software. The analysis steps on ImageJ were as follows: Bio-format importer > select file > analyze > set measurement > select mean grey value only > analyze > tools > ROI manager > add > select background (sections with no cells) > 20 cells per frame were selected for analysis using the oval shape (the video was fast tracked to the time were Ionomycin was added since that was when the cells were more visible) > multi measure > measure all slices and save excel sheet generated. Three background or blank controls (which where regions with no cells) were also included in the analysis. The average of these background controls was then subtracted from the signals and raw data for each cell at the different time points was blank corrected by subtracting each value at each time point by the average of the three background/blank controls. These blank corrected values were subtracted from the average of the baseline values and then converted to percentage by dividing each value by the signals as a percentage of the highest signal obtained in response to the Ionomycin value and multiplying by 100. The experiment was repeated three times independently. The graphs were created in Microsoft Excel. The maximum amplitude of calcium released was calculated by taking the average of the maximum amount of calcium released after the addition of KP-10, for the 20 cells that were selected per cell line. The time of maximum amplitude was calculated by taking the average of the time in which the maximum amplitude was attained for the 20 cells in each independent experiment per cell line.

### 4.6. Cell Migration Assays

#### 4.6.1. Wound-Healing Assay

An in vitro scratch assay was used to assess cell migration. BT-20 or MDA-MB-231 cells were seeded in 12-well plates at densities of 4 × 10^5^ cells/mL in Complete DMEM or at 2 × 10^5^ cells/mL in Complete DMEM-F12 media, respectively, and were incubated at 37 °C with 5% CO_2_ in a humidified incubator. The following day when the cells reached 80% confluence, the cell monolayer was scratched with a 200 µL micropipette tip across the centre of the well. Another straight line was scratched perpendicular to the first line to create a cross on each well. After scratching, the cells were washed once with PBS to remove the detached cells. Thereafter, the complete media was replaced with serum-free media containing either 0.2% propylene glycol (VC, vehicle control) or 100 nM KP-10 and were incubated for 18 h at 37 °C with 5% CO_2_ in a humidified incubator. Images of the scratches were taken using at a magnification of 20× using an Axiovert inverted microscope (Zeiss, Oberkochen, Germany) at 0 and 18 h after treatment, and the width of the scratch was measured at each time point (0 or 18 h) using ImageJ software. Migration was assessed by measuring the distance between the scraped edges on both sides at 0 h and measuring the distance migrated by the cells at 18 h for each treatment. Migration was expressed as the percentage closure of the scratch and was calculated by dividing the change in the area of the gap (area at time point 0–area at time point 18 h) by the area at time point 0 and multiplying by 100. This experiment was performed 3 times independently.

#### 4.6.2. Oris™ Migration Assay

The effect of KP-10 on MDA-MB-231 cell migration was also assessed using an Oris™ migration assay kit (Platypus technologies, Fitchburg, WI, USA), following the manufacturer’s protocol. Briefly, MDA-MB-231 cells were removed from culture flasks and the cell suspension was stained with 2.5 µM DiI (Invitrogen, Waltham, MA, USA) and incubated for 30 min at 37 °C in a humidified incubator. Stained cells were then plated into 96-well black-walled plates from Thermo Scientific (Waltham, MA, USA) with stoppers at a density of 4 × 10^4^ cells/well (100 µL per well) and were incubated overnight at 37 °C in a humidified incubator. The following day, the stoppers were removed and the media was replaced with phenol-red-free (to avoid interference with fluorescence measurement) and serum-free MEM media (Invitrogen, Waltham, MA, USA) containing either 0.1% DMSO (vehicle control; VC), 100 nM KP-10, or 10 µM BAPTA-AM + 100 nM KP-10. Wells containing media only (with no cells) were included and used for blank correction. The detection mask was inserted at the bottom of the plate and the plate was transferred to a FluoSTAR Omega microplate reader (BMG Labtech, Ortenberg, Germany) in an incubation chamber at 37 °C with 5% CO_2_. Migration was measured in real-time by measurement of fluorescence with excitation at 544 nm and emission at 590 nm. Readings were taken at 20 min intervals for 18 h using the bottom optics. Data analysis was performed using MARS data analysis software (BMG Labtech, Ortenberg, Germany). The raw data were first corrected to subtract the blank readings. Graphs were then smoothed using the moving average function of the software and the software made a calculation of the slope of linear regression. The experiment was performed three times independently with at least three technical repeats per experiment.

### 4.7. Cell Proliferation Assay

The effect of KP-10 on cell proliferation was assessed through a resazurin reduction assay. BT-20 and MDA-MB-231 cells were seeded at densities of 7.5 × 10^4^/mL or 2.5 × 10^4^/mL, respectively, in 96-well tissue culture plates (Greiner Bio-one, Kremsmunster, Austria) and were incubated overnight in a humidified incubator at 37 °C and 5% CO_2_. Differing seeding densities were used to accommodate differences in size and cell doubling times (BT-20 = 48 h, MDA = 28 h). The following day, the media was replaced with phenol-red-free (to avoid interference with fluorescence measurement), serum-free media containing 0.1% DMSO (vehicle control; VC), or 100 nM KP-10. The cells were then incubated in a humidified incubator at 37 °C and 5% CO_2_ for 24, 48, 72, 96, or 120 h after treatment. The media with the treatments were replaced daily to avoid depletion. At the end of each time point, the media was removed and replaced with media containing 0.5 mg/mL of resazurin, and cells were incubated at 37 °C for 4 h. Blank controls, which contained only the resazurin dye (no cells), were included. Fluorescence intensity was then measured using the FluoSTAR Omega microplate reader, using the bottom optics with a 530 nm excitation wavelength and a 590 nm emission wavelength. Data analysis was performed using the MARS data analysis software. The raw data were corrected by subtraction of the blank values and the standard error of the means were calculated for each timepoint and were compared to the blank. The experiment was repeated three times independently. The values obtained after blank subtraction were used to create an average and the values were used to plot a graph of time versus fluorescence intensity in Microsoft Excel. The experiment was performed three times independently with at least four technical repeats per experiment and the average blank corrected values were used to plot a graph of time vs. relative fluorescence units (RFU), in Microsoft Excel.

### 4.8. Statistical Analysis

A *p*-value of ≤0.05 was considered statistically significant. A Student’s paired *t*-test was used to compare the difference between two groups and a one-way ANOVA with Dunnett’s post-hoc test was used to compare the difference between three or more groups to a control group. Data are represented as the average ± standard error of the mean (SEM). All statistical analysis was performed using Microsoft Excel.

## Figures and Tables

**Figure 1 ijms-23-08744-f001:**
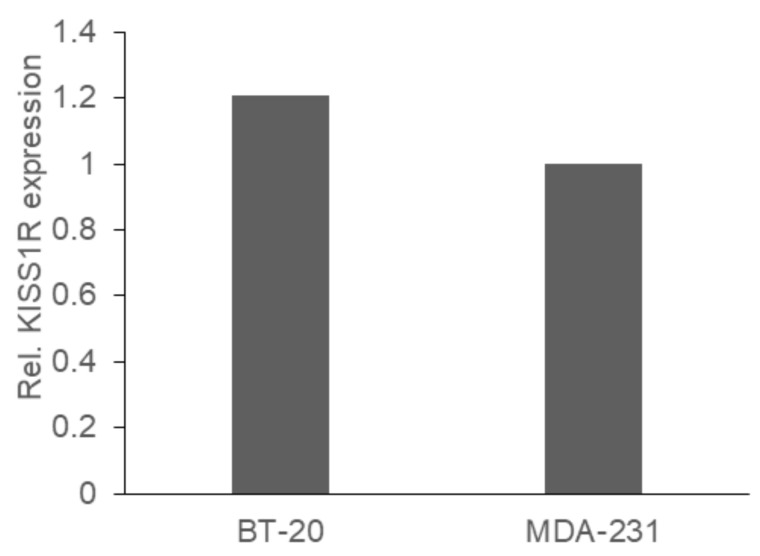
*KISS1R mRNA is expressed in BT-20 and MDA-MB-231 cells.* Total RNA was extracted from BT-20 and MDA-MB-231 cells and the mRNA was converted into cDNA using random hexamers. Quantitative real-time PCR was performed using SYBR green. The data were quantified with β-actin used as the housekeeping control. The graph represents the means from a single experiment, with three technical repeats.

**Figure 2 ijms-23-08744-f002:**
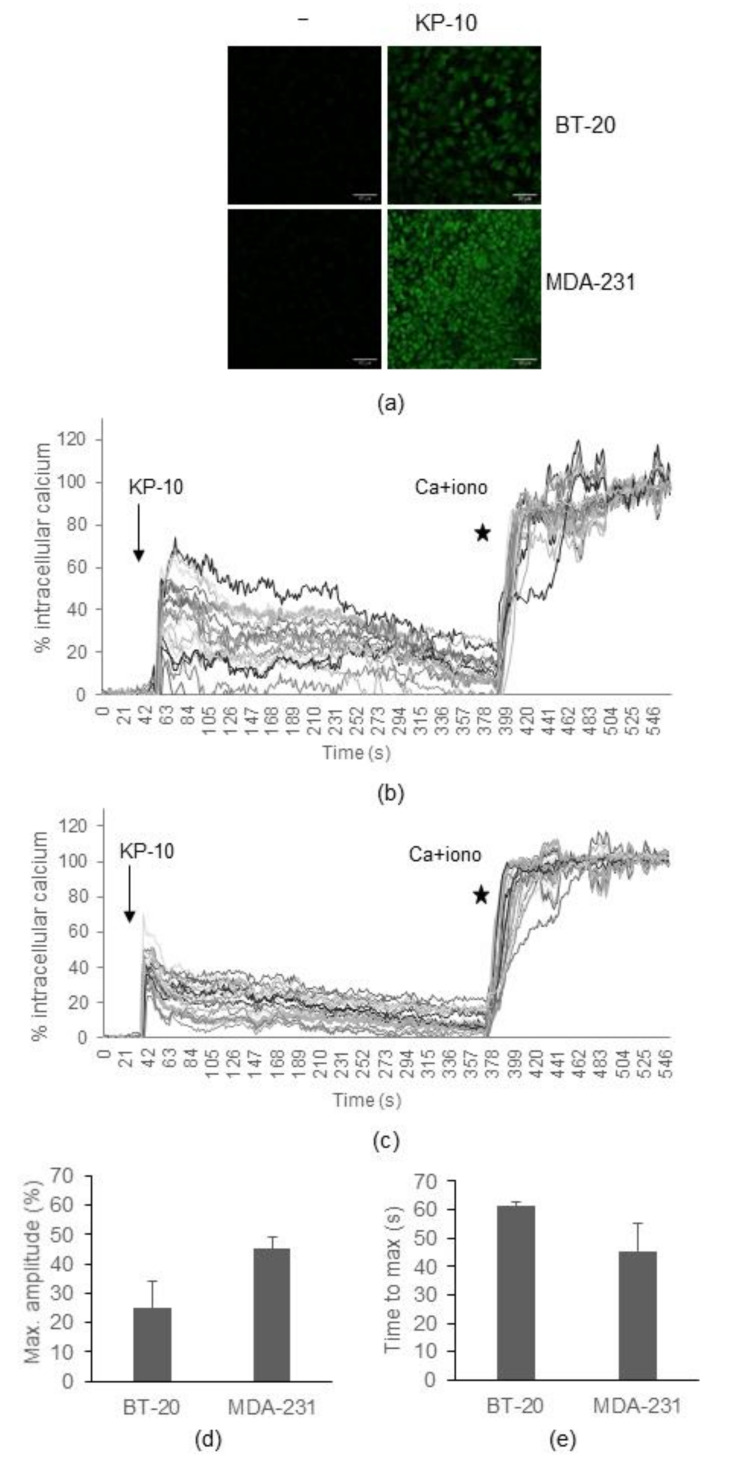
*KP-10 induces similar levels of calcium mobilization in BT-20 and MDA-MB-231 cells.* BT-20 and MDA-MB-231 cells were loaded with the calcium indicator dye, Fluo-3 AM, imaged using a Zeiss LSM800 at 20× magnification every 1.5 s before 100 nM KP-10 was added. After at least 300 s, 1 µM Ionomycin + 1 M CaCl_2_ was added (positive control) to determine the maximal Fluo-3 excitation. (**a**) Fluorescence images of BT-20 and MDA-MB-231 cells before (-) and after KP-10 stimulation. (**b**) A representative experiment showing 20 BT-20 cells tracked for fluorescence intensity before and after KP-10 stimulation (arrow) and Ionomycin stimulation (star). (**c**) A representative experiment showing 20 MDA-MB-231 cells tracked for fluorescence intensity before and after KP-10 stimulation (arrow) and Ionomycin stimulation (star). (**d**) Quantification (mean ± SEM) of the maximum amplitude reached after KP-10 stimulation in BT-20 and MDA-MB-231 cells (**e**) Quantification (mean ± SEM) of the time taken to reach maximum mobilization after KP-10 stimulation in BT-20 and MDA-MB-231 cells. There was no significant difference (*p* > 0.05) in maximum amplitude or time to maximum amplitude between the two cell lines.

**Figure 3 ijms-23-08744-f003:**
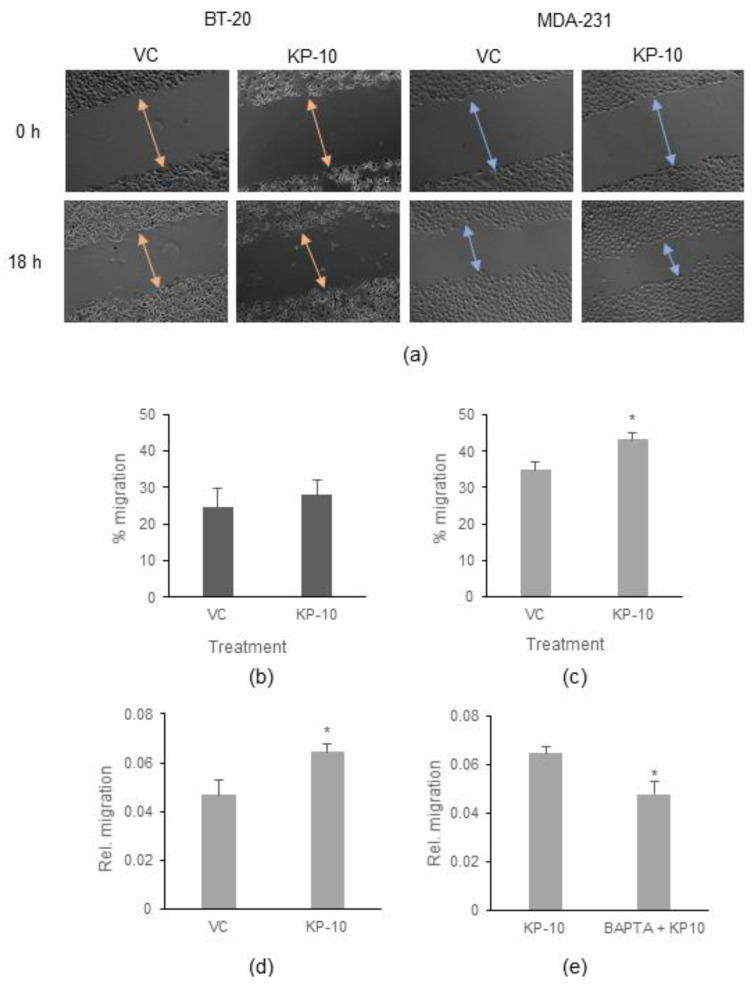
*KP-10 increases migration in a calcium dependent manner only in MDA-MB-231 cells.* (**a**–**c**) Cell migration was assessed in serum-free culture conditions via scratch assay with BT-20 and MDA-MB-231 cells treated with of 100 nM KP-10 or the vehicle control (VC). (**a**) Phase-contrast images were generated with a 5× magnification at 0 h and 18 h after stimulation. Scale bar, 100 µm. (**b**) Quantification (mean ± SEM of three independent experiments) of percentage wound closure in BT-20 cells. No significant difference (*p* > 0.05, Student’s *t*-test) between VC and KP-10 treatments. (**c**) Quantification (mean ± SEM of three independent experiments) of percentage wound closure in MDA-MB-231 cells. * *p* < 0.05 (Student’s *t*-test) for comparison of VC and KP-10 treatments. (**d**) Cell migration was measured in MDA-MB-231 cells using an Oris™ migration assay kit under serum-free culture conditions 18 h following stimulation with 100 nM KP-10 or the vehicle control (VC). Graph depicts quantification (mean ± SEM of three independent experiments) of relative migration. * *p* < 0.05 (Student’s *t*-test) for comparison of VC and KP-10 treatments. (**e**) Cell migration was measured in MDA-MB-231 cells using an Oris™ migration assay kit under serum-free culture conditions for 18 h following stimulation with 100 nM KP-10 in the presence and absence of 10 µM BAPTA-AM. The graph depicts quantification (mean ± SEM) of three independent experiments. * *p* < 0.05 (Student’s *t*-test) for comparison of presence and absence of BAPTA-AM.

**Figure 4 ijms-23-08744-f004:**
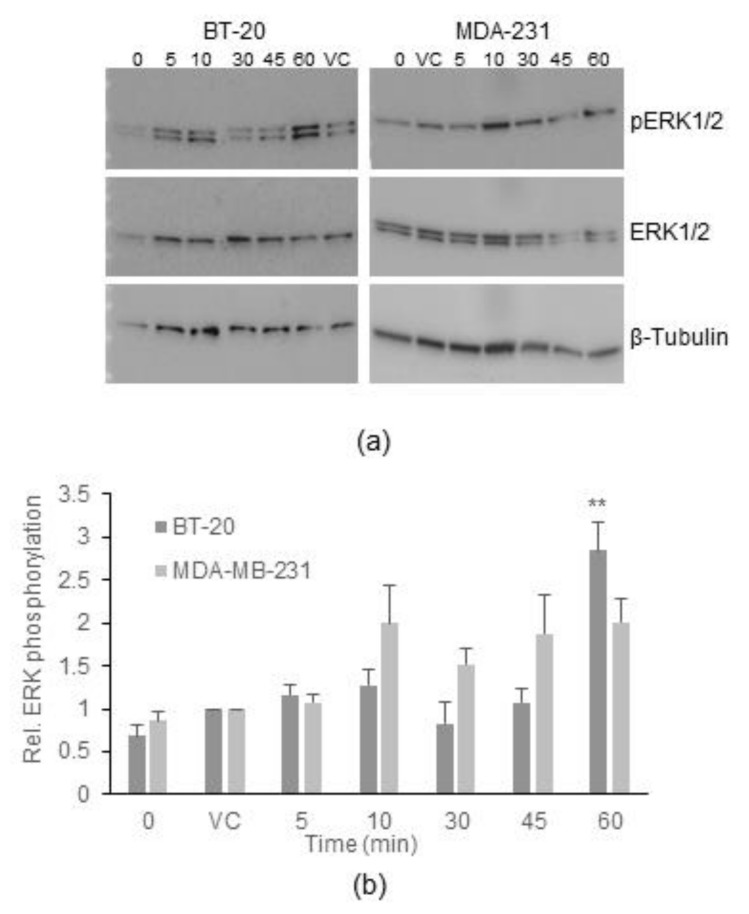
*KP-10 only activates ERK1/2 in BT-20, but not MDA-MB-231 cells.* BT-20 and MDA-MB-231 cells were serum deprived for 4 h before exposure to 100 nM KP-10 for 5, 10, 30, 45, or 60 min. Unstimulated (0 min) and vehicle treated (VC) controls were included. (**a**) A representative Western blot image depicting phosphorylated ERK1/2, total ERK1/2 and β-tubulin expression in unstimulated (0 min), vehicle control (VC), 5 min KP-10, (5), 10 min KP-10 (10), 30 min KP-10 (30), 45 min KP-10 (45), and 60 min KP-10 (60) samples. (**b**) Quantification of three independent repeats (mean ± SEM) of relative ERK1/2 phosphorylation in BT-20 (dark grey bar) and MDA-MB-231 (light grey bar) cells. ** *p* ≤ 0.01 (one-way ANOVA followed by Dunnett’s post-hoc test) to determine significant increases in ERK1/2 phosphorylation compared to the VC for each cell line.

**Figure 5 ijms-23-08744-f005:**
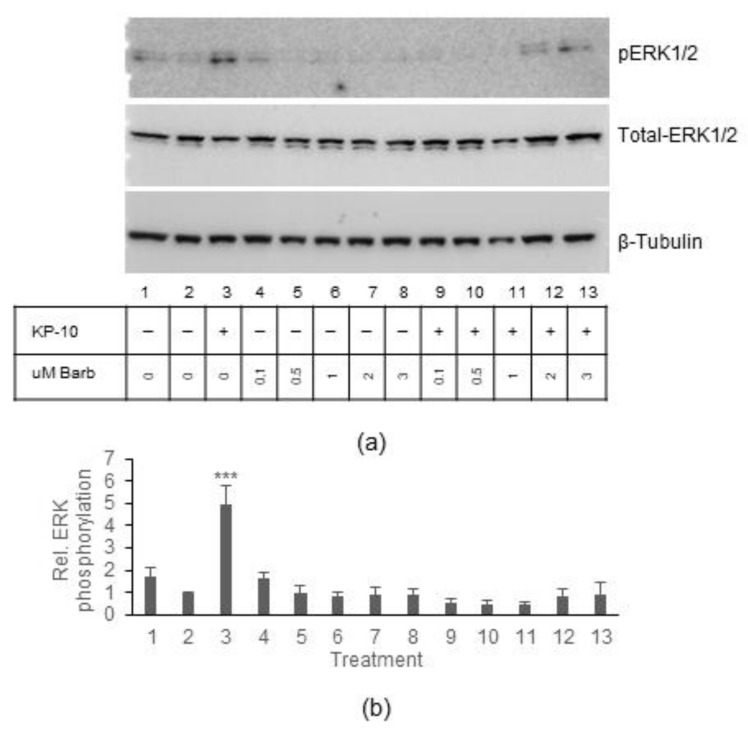
*ERK1/2 is activated in a β-arrestin1/2-dependent manner in BT-20 cells.* BT-20 cells were serum starved for 4 h and pretreated for 30 min at 37 °C in the presence or absence of 0.1 µM, 0.5 µM, 1 µM, 2 µM, and 3 µM Barbadin before stimulation with 100 nM KP-10 for 60 min. Unstimulated and vehicle controls were included. Cells were lysed with RIPA lysis buffer, separated on SDS-PAGE gels, and ERK1/2 phosphorylation was assessed through Western blotting. (**a**) A representative Western blot image depicting phosphorylated ERK1/2, total ERK1/2, and β-tubulin expression (housekeeping control) in unstimulated (1), vehicle control (2), 100 nM KP-10 only (3), 0.1 µM (4), 0.5 µM (5), 1 µM (6), 2 µM (7), 3 µM Barbadin (8), 0.1 µM Barbadin + KP-10 (9), 0.5 µM Barbadin + KP-10 (10), 1 µM Barbadin + KP-10 (11), 2 µM Barbadin + KP-10 (12), and 3 µM Barbadin + KP-10 (13) samples. (**b**) Quantification of four independent repeats (mean ± SEM) of relative ERK1/2 phosphorylation *** *p* ≤ 0.001 (one-way ANOVA followed by Dunnett’s post-hoc test) to determine significant increases in ERK1/2 phosphorylation compared to VC (2).

**Figure 6 ijms-23-08744-f006:**
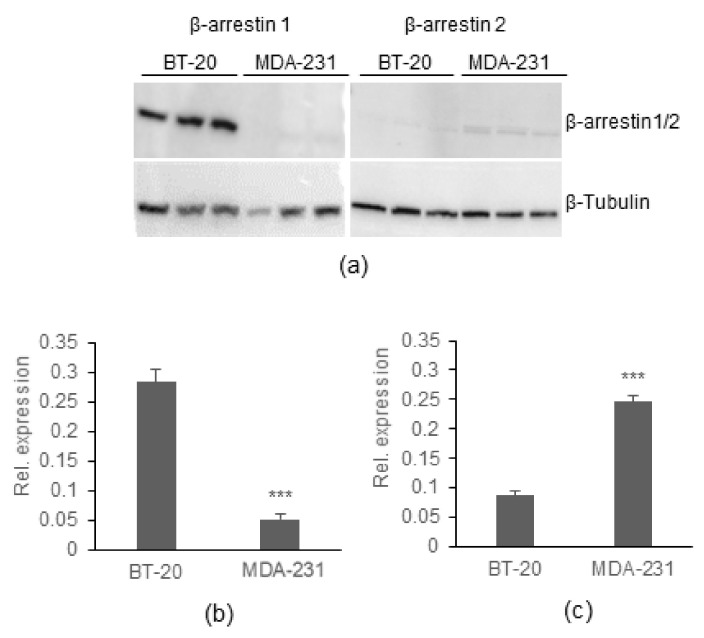
*β-arrestin 1 and 2 are differentially expressed in BT-20 and MDA-MB-231 cells.* Three independent batches of BT-20 and MDA-MB-231 cells were collected and analysed for β-arrestin 1 and 2 protein expression by Western blot. (**a**) Western blot image depicting β-arrestin 1, β-arrestin 2, and β-tubulin (housekeeping control) expression in BT-20 and MDA-MB-231 cells (**b**) Quantification (mean ± SEM) of the relative β-arrestin 1 expression (**c**) Quantification (mean ± SEM) of the relative β-arrestin 2 expression. *** *p* ≤ 0.001 (Student’s *t*-test) to compare expression of β-arrestin 1/2 in the different cell lines.

**Figure 7 ijms-23-08744-f007:**
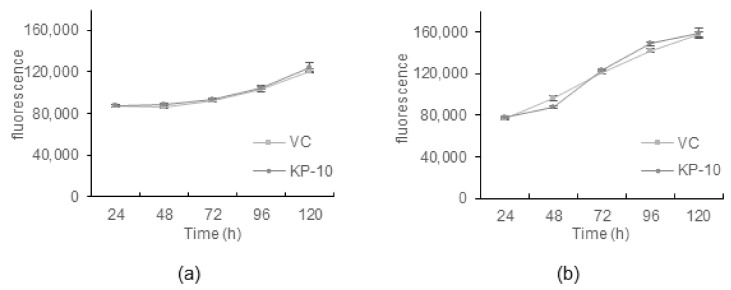
*KP-10 has no effect on the proliferation of BT-20 or MDA-MB-231 cells.* Proliferation of (**a**) BT-20 and (**b**) MDA-MB-231 cells was measured by resazurin conversion assay after 24, 48, 72, 96, and 120 h treatment in the presence of 100 nM KP-10 or the vehicle control (VC) under serum-free culture conditions. Data are presented as the mean ± SEM from three independent assays.

## Data Availability

All data recorded in this study is archived online within Benchling.com and is available upon request to the corresponding author.

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
