# Peer review of "Lack of Oestrogen Receptor Expression in Breast Cancer Cells Does Not Correlate with Kisspeptin Signalling and Migration"

_ijms, 2022, doi:10.3390/ijms23158744_

Round 1
Reviewer 1 Report
Please check the attached review report.

Author Response
Thank you for your thoughtful comments. We have read and addressed all comments included in the associated file. Please find our response to the comments below and attached as a doc file.
- Since authors studied Kisspeptin-10 and there is more than one type of it, it must be specified (KP-10) wherever it could be starting from the title
As kisspeptin is the collective term for all forms, we follow the convention to use Kisspeptin for more general references, such as in the title, as this points to the signalling brought about by Kisspeptin. However, we take note of the reviewer's concerns and, thus, we have rationalised the use of the two terms in the revised manuscripts and use KP-10 where we speak of our experiments and use kisspeptin where we speak of the system in general or from literature
- Authors used different VC controls (0.1% DMSO or 0.2% propylene glycol) in cell proliferation assays and cell migration assays, respectively, why? How was KP-10 prepared? Why did the authors use 100nM KP-10, and what the volume was to get 100nM KP-10? What would be happened if different KP-10 concentrations were used?
100 nM was selected as it is commonly used by others. Indeed, a concentration curve of ERK activation in bt20 cells was performed as a prelude to the experiments presented and demonstrated that, in this system, this is the minimum concentration at which an ERK response is seen. To clarify this, a sentence has been added to the revised manuscript as follows.
‘Previous studies have shown that 100 nM KP-10 induces the migration of MDA-MB-231 cells’
M and M also adjusted as follows:
Originally, KP-10 was solubilized in 0.2% propylene glycol and was used in the first migration assays (scratch assays). Subsequent preparations of KP-10 were prepared in 0.1% DMSO such that, for all assays other than the scratch assay, the vehicle control was 0.1% DMSO.
Indeed, it would be interesting to examine dose-dependent effects of KP-10.
- 1 does not have error bars. Why?
This graph represents a single biological repeat with three technical repeats.
Sentence has been added to legend to clarify
- Page 3, line 115; it was stated that “Cells were cultured overnight in 8-well incubator slides” while it was stated in the materials and methods section that cells were cultured in 8-well microslide plate. Which one is correct?
Text has been corrected to microslide plates in line 115
- Page 4, lines130-133; “A representative experiment showing 20 BT-20 cells tracked …… A representative experiment showing a number of MDA-MB-231 cells tracked for fluorescence intensity …” Why were twenty BT-20 cells tracked versus a nonspecific number of MDA-MB-231 cells?
This graph also represents 20 cells from MDA. Corrected in text
- Since Fig.2 (b and c) have big free spaces, writing KT-10 and Ionomycin + CaCl2 above the arrow and star, respectively, is highly recommended. That would make the figure easier to be read.
Change has been made
- Having images for Fig.3 (d and e) would be helpful.
Since these are results from the Oris migration assay rather than scratch assay, there are no associated images
- Page 5, line 165; “MDA-MB-231 cells treated with of 100 nM KP-10 or vehicle control (VC, 0.1% propylene glycol).” Please, confirm the propylene glycol concentration.
This was a typo. It has been corrected to 0.2%
- Page 6, lines 190-193; “Multiple biological repeats were quantified and statistical analysis indicated that KP-10 induced ERK phosphorylation only at 60 min in the BT-20 cells and not at all in the MDA-MB-231 cells.” Please, rewrite this statement. There was a difference regarding the MDA-MB-231 cells but it was not statistically significant.
Text altered to say: ‘Statistical analysis of multiple biological repeats indicates that KP-10 induced ERK phosphorylation at 60 min in the BT-20 cells. In the MDA-MB-231 cells, there was a smaller increase in ERK phosphorylation from 10 minutes, but this was not statistically significant.’
- 4a. There are 2 bands of pERK1/2 in BT-20 blot while it is one band in MDA-MB-231 blot. However, it is the opposite situation for ERK1/2. Why?
If one looks closely a second band is visible in the bt20 total erk blot. It is just below the darker band. It is quite common to see differences in phosphor ERK bands with some cells showing both and others only showing one. Why this happens we do not know
- From Fig.4b. it seems that the relative ERK phosphorylation in MDA-MB-231 at 10 and 60 min was much higher compared to unstimulated (0 min) and vehicle control (VC). Please, verify the statistical analysis.
We have repeated this experiment five times to verify the erk phsophorylation status and have not found significant differences. We do see variation in blots.
- Page 8, lines 236-237; “Conversely, b-arrestin 2 was expressed 3,5-fold higher in the MDA-MB-231 cells…” it should be “Conversely, -arrestin 2 was expressed 3.5-fold higher in the MDA-MB-231 cells…”.
Corrected in text
- Page 10, lines 312-314; “Calcium mobilization after stimulation with KP-10 increased…. to max mobilization were similar in both cell lines.” Please, rewrite this sentence considering the results on page 3, lines 120-122.
Sentence changed to: ‘BT-20 cells reached an average maximum level of calcium of 25.2% of that elicited in the positive control after 61.5 s while MDA-MB-231 cells reached a maximum of 45%, 45.5 s after stimulation. However, there was some cell-to-cell variability and overall, there was no significant difference in amplitude between the cell lines. There was also no difference in time to maximum amplitude’

Reviewer 2 Report
This is an interesting and well-written manuscript that attempts to resolve conflicting data about the roles of kisspeptin and its receptor, KISS1R, in estrogen receptor negative (ER-) breast cancer. Kisspeptin and its receptor have been identified as potentially involved in metastasis specifically in ER- breast cancers, though much of this previous data has used overexpression of KISS1R in cell lines, a relatively artificial system. Previous studies have suggested that breast cancer subtype determines the effects of kisspeptin signaling. The authors hypothesize that different ER- cell lines could respond to kisspeptin in a context-dependent manner and use MDA-231 and BT-20 lines to study the effects of kisspeptin stimulation on proliferation, migration, invasion, and calcium signaling. Their conclusion is that not all ER- breast cancer cells respond in the same way and therefore additional studies are needed to identify cells that respond detrimentally to kisspeptin signaling. The manuscript is interesting, will advance the field, and the conclusions are well-supported. Some major and minor concerns should be addressed prior to acceptance.
Major concern
I would like to see evidence of protein expression of KISS1R in these cell lines. It’s well-established that the presence of mRNA does not necessarily correlate to protein expression, and since the paper hinges on the presence of KISS1R protein, it is important to see evidence of protein in both these cell lines.
Minor concerns
1. I’m not sure if the low resolution of some of the figures is an artifact of the journal’s system or if the resolution is directly due to the figures, but Fig 1, 2A, 6, and 7 are fuzzy and hard to read.
a. Fig 7 is also very hard to distinguish between treatments
2. There’s a figure typo in line 144 (should be “Figure 3A”)
3. Figure 4a: the legend says the blots were probed using MAPK antibodies, but the blot is labeled ERK1/2.
Author Response
Thank you for your comments. We have read these and adjusted our manuscript accordingly.
Specifically we have altered the following:
In response to the major concern regarding KISS1R protein expression the following:
We have spent considerable time and effort attempting to obtain a conclusive answer wrt the protein expression of KISS1R in these cell lines. We obtained a custom antibody as well as a commercial antibody which we tested on cells overexpressing a FLAG tagged KISS1R recombinant protein and found that the KISS1R antibodies did not recognise the same protein as a FLAG antibody despite numerous optimisation attempts. Western blot analysis of several cell lines beyond the ones published here also did not provide evidence that either antibody was able to detect endogenous KISS1R. This led us to test calcium mobilisation as a way to determine if functional KISS1R protein was present and responsive to KP-10. We believe that the assay does indeed confirm the RT-PCR data suggesting that KISS1R mRNA is expressed and results in the production of functional KISS1R protein.
To clarify this we added the following to the results section of the RT-PCR: 'Endogenous KISS1R protein expression was also assessed by Western blot (data not shown) using both a custom generated and commercially available antibody, but control experiments were unable to produce convincing evidence that either specifically detected KISS1R. Therefore, we chose to assess KP-10-dependent signaling to confirm expression of functional KISS1R in these cell lines.'
Minor comments
- There were indeed some low res images. Apologies. We have adjusted and reintroduced these figures as high res images. Fig 7A has also been improved to make different treatments easily recognizable.
- Typo has been corrected
- The MAPK was added in error and has been removed.
Round 2
Reviewer 2 Report
The revisions are acceptable and thank you for the additional clarifications in the manuscript; we've had the same experience with antibodies to a different protein and completely understand.